# A Case Study Demonstrates That the Litter of the Rare Species *Cinnamomum migao* Composed of Different Tissues Can Affect the Chemical Properties and Microbial Community Diversity in Topsoil

**DOI:** 10.3390/microorganisms10061125

**Published:** 2022-05-30

**Authors:** Yuangui Xie, Xiaofeng Liao, Jiming Liu, Jingzhong Chen

**Affiliations:** 1College of Forestry, Guizhou University, Guiyang 550025, China; yuangui_xie@163.com (Y.X.); gs.chenjz20@gzu.edu.cn (J.C.); 2Subordinate Departments of the Academy, Guizhou Academy of Sciences, Guiyang 550025, China; 3Guizhou Institute of Mountain Resources, Guizhou Academy of Sciences, Guiyang 550025, China; lxfnsd@163.com

**Keywords:** *Cinnamomum migao*, forest litters, soil microbiomes, nutrient return

## Abstract

The decomposition of litter plays an important role in the return of forest soil nutrients, as well as the growth and productivity of plants. With this study, we aimed to determine the impact of litter mulching on different tissues of *Cinnamomum migao*, a rare Chinese endemic species. In particular, seeds and pericarp are easily overlooked components of *C. migao* litter. In this study, we tested control (uncovered litter) and litter (leaf, branch, seed, and pericarp) mulching conditions and conducted a one-year litter decomposition experiment. The enzyme activities of urease enzyme (UE) and invertase enzyme (INV) were significantly improved by litter mulching. Catalase (CAT) enzyme activities in leaf, branch, and seed litter mulching were lower than in the control, whereas CAT activity in pericarp mulching was significantly higher than in the control. Although *Mortierella, Cladophialophora*, *Acidothermus*, *Sphingomonas*, and *Burkholderia* were the dominant microbes of topsoil in different mulching treatments, there were differences in the number and connectivity of microbial communities, and this change was correlated with soil organic carbon (SOC) and CAT enzyme activity. Compared with leaves and branches, seeds and pericarp as litter are also very important for nutrient return and affect topsoil microbes in *C. migao* forest, which may be of significance for the growth feedback of *C. migao* in biennial bearing.

## 1. Introduction

Litter is a crucial pathway of nutrient return and material flow in forest ecosystems. Litter decomposition can account for 70–80% of forest N, P, and K supply in material return [1]. In general, litter decomposition is usually driven by local climate and is mainly affected by local litter types, soil biota, and abiotic environmental factors [2]. Soil microbes play a key role in litter decomposition. Some specific soil microorganisms often participate in and dominate the process of litter decomposition and nutrient return [3].

Depending on the source of litter, there are considerable differences in its physical structure and chemical composition. For example, the phenolic compounds in some litters affect the colonization time and decomposition rate of microorganisms near the litter. The degradation rate of complex compounds accelerates when litter attracts certain soil microorganisms that degrade lignin to colonize [4]. The soil microbial community in litter mulch recruits specific soil microbes to colonize due to the specificity of the nutrient supply and leaching chemical components of different litters. Changes in the input type of leaf litter can change the soil microbial community and nutrient dynamics, as well as the soil organic carbon dynamics. These factors have a considerable impact on the physical and chemical properties of the soil in a forest ecosystem and the soil microbial community [5]. Some secondary metabolic processes in litter can permeate the soil through rain leaching and affect the process of litter decomposition and nutrient return. Although many compounds (such as alkaloids) in fresh plants metabolize rapidly in litter and have limited interaction with soil microorganisms [6], other chemicals released from litter remain in the soil after partial decomposition, which has a significant impact on the composition and structure of the soil microbial community. For example, flavonoids can significantly attract Rhizobium, and soil microorganisms secrete phenol oxidase and peroxidase [7]. First, litter contains generous amounts of carbon and nitrogen, which is an important nutrient source for soil microbial life activities. Litter mulching also bring out microorganisms to secrete urease, phosphatase, and invertase enzymes to promote the mineralization of organic matter [8]. Depending on the species source and tissue type, the ability of litter to recruit and attract soil microbes in the process of litter decomposition and ultimately form a specific microbial community structure varies [9]. Dead branches and fallen leaves make up the bulk of forest litter, although for some plants that can produce a large number of fruits, fruit fall is also an important source of forest litter [10,11]. However, research on the impact of litter on soil microorganisms has mainly focused on plant leaves and branches, and little is known about the impact of pericarp and seeds on the soil microbial community and chemical properties in the decomposition process.

*Cinnamomum migao* H. W. Li, an evergreen arbor belonging to the Lauraceae family, is a rare endemic species in China. Each wild population contains only about five adult plants; as a result, this species has a low population [12]. *C. migao* fruit is a very important traditional medicine and spice in the southwest region of China. An adult plant can produce as much as 2000 kg of fruit a year [13,14]. After ripening, a large number of *C. migao* fruits fall from trees up to about 30 m tall, leading to the separation of the seed and pericarp. A small number of seeds naturally germinate to become seedlings or are eaten by animals, and a large number of seeds are decomposed by soil microbes, returning the nutrients to the soil [14]. *C. migao* exhibits the phenomenon of biennial bearing. After the first fruiting year, the amount of fruit produced the following year decreases sharply or the tree does not bear fruit at all. One theory holds that the uneven distribution of resources in the reproductive growth and vegetative growth stages, as well as the competition between different tissues of the tree for nutrients, are the causes of biennial bearing, and plants need to acquire returned nutrients from the environment to bear fruit [15,16,17]. The return of nutrients after the interaction and decomposition of a large amount of litter, including fruits, with soil microorganisms is an important pathway of nutrient return regulation [18,19]. There are differences between the chemical components and physical structures of *C. migao* branches, leaves, seeds, and pericarp. Flavonoids, alkaloids, and lignin are the main components in leaves; lignin is the main component in branches; terpenes, hydrocarbons, and other volatile oils are the main components in pericarp; and crude fat and protein are the main components in seeds. In the process of decomposition after litterfall, there must be differences in the effects on soil microbial and chemical properties [20,21]. However, the effects of litter formation of different *C. migao* tissues on topsoil microbes and chemical properties—especially the effects of a large number of fruits on soil microbes and nutrient return in the forest in the fruit-bearing years of *C. migao*—are still unknown. Therefore, we conducted an in situ experiment in a C. migao natural forest, using four tissues (leaves, branches, seeds, and pericarp) as the artificial mulching condition, with uncovered topsoil as the control, in order to explore the feedback process of the soil microbial community, soil nutrients, and important soil enzymes (S-UE, S-ACP, S-CAT, and S-INV). The aim of this study was to answer the following questions: (1) What will happen to the chemical properties of the topsoil of a *C. migao* forest soil under litter mulching composed of different tissues? (2) Does the soil microbial community structure in under *C. migao* pericarp and seed mulching have the same impact as that under branch and leaf mulching?

## 2. Materials and Methods

### 2.1. Study Site

The study site is located in Luodian County, Guizhou Province, China, 106°23′–107°03′ E, 25°04′–25°45′ N, at an altitude of 436 m. The study site is located in the slope zone of the transition between Yunnan-Guizhou Plateau and the hills, and it has a stepped terrain that is high in the north and low in the south. The annual average temperature is 20.35 °C, the average temperature in extremely cold months is 8.0–10.45 °C, the annual accumulated temperature ≥10 °C is 5750–6500 °C, the frost-free period is 335–349.5 days, the annual total sunshine hours are 1297.7–1600 h, and the annual rainfall is 1200 mm.

### 2.2. Sample Preparation and Experimental Design

In October 2018, when the *C. migao* fruit was ripe, nylon nets were set up to collect fresh *C. migao* litter that was not in contact with the ground. The litter was classified as leaves, pericarp, seeds, or dead branches. The surface litter and weeds were removed, and the topsoil in the sample area in the *C. migao* forest was evenly mixed and collected to determine the initial physical and chemical properties (Appendix A). Then, 3 × 5 m sample plots were set up. Nylon mesh bags of litter composed of different tissues were evenly arranged to mulch the sample plot (three repetitions for each treatment), and no litter was mulched for the control (three repetitions). A total of 15 sample plots were designated, and natural decomposition was carried out for 360 days. A blocking net was set above each plot to prevent litter from falling into the test area, and the litter on the net was removed regularly in a timely manner to avoid affecting the experimental treatment (Figure 1).

### 2.3. Sample Collection

In October 2019, the topsoil (0–10 cm) was collected after removing the litter on the surface. After five-point sampling and layered collection, Sterile polyethylen gloves were used to remove the litter from the sampling point. Then, we used a sterile shovel to scrape the surface impurities and a disposable sterile sampler to collect specimens from the sampling point. The topsoil was evenly mixed and passed through a 2 mm sterile grid sieve. After removing the impurities, such as soil gravel, the topsoil was loaded into a sterile polyethylene pipe for sealing. The topsoil samples were immediately stored in an icebox and brought back to the laboratory. Collected soil samples were then divided into two parts. One was dried at environmental temperature (25 °C) for determination of soil chemical properties, and the other was used for microbial high-throughput sequencing.

### 2.4. Soil Nutrient and Enzyme Concentration Analysis

Traditional soil agrochemistry was used to analyze soil properties, including total nitrogen (TN), total phosphorus (TP), total potassium (TK), alkali hydrogenated nitrogen (AN), available phosphorus (AP), available potassium (AK), and soil organic carbon (SOC) [3]. Soil acid phosphate (ACP), soil urease (UE), soil catalase (CAT), and soil invertase (INV) were detected with reference to the instructions of the reagent company (Beijing Solarbio Bioscience Technology Co., Ltd., Beijing, China).

### 2.5. Soil DNA Extraction and PCR Amplification

Microbial community genomic DNA was extracted from topsoil in different litter mulching samples using a DNeasy PowerSoil kit (100) according to the manufacturer’s instructions. The DNA extract was checked on 1% agarose gel, and DNA concentration and purity were determined with a NanoDrop 2000 UV-vis spectrophotometer (Thermo Scientific, Wilmington, DE, USA). The hypervariable region (V3–V4) of the bacterial 16S rRNA genes 338F (5′-ACTCCTACGGGAGGCAGCAG-3′) and 806R (5′-GGACTACHVGGGTWTCTAAT-3′), as well as the ITS rRNA genes ITS1F (CTTGGTCATTTAGAGGAAGTAA) and ITS2 (GCTGCGTTCTTCATCGATGC), were amplified with primer pairs by an ABI GeneAmp^®^ 9700 PCR thermocycler (ABI, Los Angeles, CA, USA). PCR amplification of the 16S rRNA gene was performed as follows: initial denaturation at 95 °C for 3 min, followed by 27 cycles of denaturing at 95 °C for 30 s, annealing at 55 °C for 30 s, extension at 72 °C for 45 s, and a single extension at 72 °C for 10 min, ending at 4 °C. The PCR mixtures contained 5 × 4 μL TransStart FastPfu buffer, 2 μL 2.5 mM dNTPs, 0.8 μL forward primer (5 μM), 0.8 μL reverse primer (5 μM), 0.4 μL TransStart FastPfu DNA polymerase, 10 ng template DNA, and ddH_2_O for a final volume of 20 μL. PCR reactions were performed in triplicate. The PCR product was extracted from 2% agarose gel, purified using an AxyPrep DNA gel extraction kit (Axygen Biosciences, Union City, CA, USA) according to the manufacturer’s instructions, and quantified using a Quantus™ Fluorometer (Promega, Madison, WI, USA).

### 2.6. Illumina MiSeq Sequencing

Purified amplicons were pooled in equimolar and paired-end sequences on an Illumina MiSeq PE300 platform/NovaSeq PE250 platform (Illumina, San Diego, CA, USA) according to the standard protocols of OE Biotech Microbe Seq Technology Co., Ltd. (Shanghai, China).

### 2.7. Statistical Analyses

#### 2.7.1. Processing of Sequencing Data

The raw gene sequencing reads were demultiplexed, quality-filtered by fastp version 0.20.0, and merged by FLASH version 1.2.7 with the following criteria [22,23]: (i) the 300 bp reads were truncated at any site with an average quality score <20 over a 50 bp sliding window, and truncated reads shorter than 50 bp were discarded, as well as reads containing ambiguous characters; (ii) only overlapping sequences longer than 10 bp were assembled according to their overlapped sequence, the maximum mismatch ratio of the overlap region was 0.2, and reads that could not be assembled were discarded; and (iii) samples were distinguished according to the barcode and primers, and the sequence direction was adjusted, with exact barcode matching and two nucleotide mismatches in primer matching. Operational taxonomic units (OTUs) with a 97% similarity cutoff^]^ were clustered using UPARSE version 7.1, and chimeric sequences were identified and removed. The taxonomy of each OTU representative sequence was analyzed by RDP Classifier version 2.2 against the 16S rRNA database using a confidence threshold of 0.7 [24].

#### 2.7.2. Variance and Diversity Composition Analysis

One way analysis of variance (ANOVA) was performed using SPSS (Chicago, IL, USA), and the Tukey significance between different treatments was tested (*p* < 0.05). Mapping was carried out using Origin 2021 (Origin Lab, Northampton, MA, USA). The alpha and beta diversity of the microbial community were analyzed using the vegan package in R, and mapping was carried out using the ggplot package. The potential impact relationship between the soil microbial community structure and the soil chemical properties was analyzed using Canoco (Microcomputer Power, 2012, Version 5).

#### 2.7.3. Network Construction and Analysis

Networks were constructed for topsoil fungal and bacterial communities based on OTU relative abundances under different litter mulching conditions, yielding a total of 10 networks. Only OTUs detected in all replicate samples were used for network construction. The microbial ecological networks were calculated and constructed using the Hmisc and igraph packages in R. For visualization and network structure analysis, Cytoscape 3.8.2 and gephi were used to analyze the network composition. According to the values of *Z* and *P*, the network nodes were defined as follows: ultraperipheral node (*Z* < 2.5, *p* < 0.05), peripheral node (*Z* < 2.5, 0.05 < *p* < 0.652), non-hub connectors (*Z* < 2.5, 0.625 < *p* < 0.8), non-hub kinless nodes (*Z* < 2.5, *p* > 0.8), provincial hubs (*Z* > 2.5, *p* > 0.75), and connector hubs (*Z* > 2.5, 0.3 < *p* < 0.75).

## 3. Results

### 3.1. Soil Biochemical Properties

After 1 year of litter decomposition, soil urease and invertase increased compared with the control group after litter mulching (except for the leaf litter treatment, for which urease and invertase levels were not significantly higher than those in the control) (*p* < 0.05) (Figure 2a). The activities of urease and invertase under pericarp mulching were the highest (Figure 2b). Soil acid phosphatase did not differ significantly under different treatments (*p* > 0.05) (Figure 2c). Catalase levels were significantly higher under pericarp mulching than in the control, and catalase levels under leaf, seed, and branch litter mulching conditions were significantly lower than those in the control (*p* < 0.05) (Figure 2d).

AN was significantly higher than in the control under branch, leaf, and pericarp mulch, whereas TN was significantly higher than under the control, as well as leaf and branch mulch. AP litter mulching was significantly higher than that in the control, and the content of pericarp and branch treatment was the highest (Figure 3c). There was no difference between the TP of leaf mulching and the control, and pericarp, seed, and branch litter mulching TP was significantly higher than that of the control. The TP content of was the highest under the pericarp mulching condition (Figure 3d). AK in the litter mulching condition was significantly higher than that in the control, and the AK content of pericarp mulching was the highest (Figure 3e). The content of TK under litter mulching was higher than that under branch mulching, and the other litter types resulted in slightly higher TK contents than that in the control (Figure 3f). SOC increased in litter mulching and was significantly higher than that in the control, except for under branch mulching (Figure 3g).

### 3.2. Alpha Diversity and Community Composition

The Chao1 index of topsoil was higher than that of the control for all types of litter. Similarly, the Shannon index was lower under pericarp mulching than that of the control, and the Shannon index values of the leaf, seed, and branch mulching treatments were higher than that of the control group, although there was no significant difference (*p* > 0.05), and the Simpson index had no difference compared with the control (Table 1).

Regarding the microbial composition of litter in the *C. migao* forest, the fungi groups mainly comprised Mortierella, Cladophialophora, Geminibasidium, and Umbelopsis (Figure 4a); and main bacteria were Acidothermus, Sphingomonas, and Burkholderia (Figure 4b).

To further clarify the species comprising the microbial community in the topsoil of the *C. migao* forest under all treatments, LEfSe analysis was carried out. In the fungal microbiome, there were significant differences between 35 species in the unmulched treatment (control), to which Mycenaceae, Chytridiomycota, and Hypocreaceae contributed the most. There were significant differences between 25 species in the leaf mulching condition, to which Fusidium, Rozelomycota, and Chlorocboria made the maximum contribution. There were significant differences between 31 species in the branch mulching condition, to which Sordariales, Lophistoma, and Lasiosphaeriaceae made the maximum contribution. There were significant differences between 48 species in the pericarp mulching condition, to which Chaethyriales, Tremellomycetes, and Tremellales contributed the most. Seed mulching had 17 species that were significantly different, with Mortierellomycotina_cls_Incertae_sedis, Mortierellaceae, and Zygomycota showing the most conspicuous differences (Figure 5a and Appendix A). The differences in bacterial community composition in topsoil under different treatments were as follows. There were significant differences between 37 species in the unmulched treatment (control), with Gemmatimonadaceae and Gemmatimonadetes making the highest contribution. There were significant differences between 21 species in the leaf mulching condition, with GR_WP33_30, Chryseobacterium, and Telmatospirillum contributing the most to these differences. There were significant differences between 20 species in the branch mulching condition, with Pantoea, *Arachis hypogaea* var. *vulgaris*, and Sulfuritalea making the most significant contribution. There were 34 significantly different species in the pericarp mulching condition, including Nevskiaceae and Hydrocarboniphaga. There were 28 species that were significantly different in the seed mulching condition, including Gammaproteobacteria, Pseudomonadaes, and Pseudomonadaceae (Figure 5b and Appendix A).

### 3.3. Beta Diversity Analysis

Principal component analysis (PCA) showed that for the fungal community, the contribution of PCA1 was 19.15%, and that of PCA2 was 12.82% (Figure 6a). The diversity of the fungal community in the topsoil of the *C. migao* forest under control and pericarp mulching was significantly different. The fungal community diversity under seed, branch, and leaf mulching clustered together, which did not reflect the obvious diversity difference. For the bacterial community, the contribution of PCA1 was 21.97%, and that of PCA2 was 12.93% (Figure 6b).

There was a significant difference between the control and litter mulching. The repetitions were similar in pericarp and leaf mulching, but there was no significant aggregation in seed and branch mulching. To further test the differences in microbial diversity in the topsoil under the *C. migao* forest in all treatments, the non-metric multidimensional scaling (NMDS) analysis showed that the fungal sorting result was better (stress = 0.054), and the control obviously gathered together, which was significantly different from the results under other litter mulching treatments. The similarity between pericarp and leaf mulching was higher, whereas there was less similarity between branch and seed mulching (Figure 7a). The bacterial sorting results were very representative (stress = 0.048), and the control was obviously clustered together, which was significantly different in litter mulching. However, of the litter mulching treatments, only pericarp mulching had a higher similarity, whereas the other treatments were more dispersed (Figure 7b).

### 3.4. Microbial Ecological Network Construction and Correlation Analysis of Soil Chemical Properties in Different Ttreatments

We constructed 10 ecological networks of fungi and bacteria in the topsoil mulched with different treatments (Figure 8). The number of nodes, number of edges, and number of neighbors of the microbial ecological network show that the bacterial community ecological network structure under all treatments in the topsoil in *C. migao* forest is more complex and compact than that of the fungal community (Appendix A).

In particular, the ecological network structure of both the fungal community and bacterial community in topsoil under the pericarp treatment was the most complex. In the composition of network nodes, the peripheral node was the node in both the fungal and bacterial community ecological networks that made up the main constituent node of the ecological network (more than 70%) in the leaf and branch treatments. There was no connector hub in the fungal ecological network under the control treatment, whereas other litter treatments had a connector hub. Bacterial ecological network connector hubs do not exist under pericarp mulching (Table 2). In addition to the similarity of network structure, the community structure of topsoil in the seed and control litter mulching conditions was relatively similar. The community structure of topsoil in the branch and leaf litter mulching conditions was also relatively similar, whereas the community structure of topsoil in the pericarp mulching condition was different from that under seed, leaf, and branch litter mulching (Appendix A).

Mycenaceae, Hypocreaceae (Figure 9a), Gemmatimonadaceae, and Gemmatimonadetes (Figure 10a) had the greatest impact on nutrients in the topsoil in the control. Fusidium, Chlorocboria (Figure 9b), uncultured_Syntrophobacterales_bacterium, and GR_WP33_30 (Figure 10b) had the greatest impact on nutrients. In the topsoil in the branch mulching condition, Lophistoma, Sordariales (Figure 9c), Pantoea, and *A. hypogaea* var. *vulgaris* (Figure 10c) had the greatest impact on nutrients. Under pericarp mulching, Chaetotyriales, Tremellales (Figure 9d), and Nevskiaceae (Figure 10d) had the greatest impact on nutrients. Mortierellalaceae, Trichocomaceae (Figure 9e), Gammaproteobacteria, and Pseudomonadales (Figure 10e) had the greatest impact on soil nutrients in seed mulching. In general, the composition of the microbial community was most affected by SOC and CAT in all treatments.

## 4. Discussion

### 4.1. Effects of Litter Mulching with Different C. migao Tissues on the Chemical Properties of Topsoil

Litter plays an important role in the nutrient return of forest ecosystems, which is closely related to the primary productivity of ecosystems. Litter mulching improves soil nutrients and changes soil enzyme activity [25,26]. In our study, although the TP in pericarp mulching was slightly lower than that of the unmulched litter (control), leaf, seed, and branch mulching increased the input of soil nutrients, which was basically consistent with the changes in soil nutrients under litter decomposition. It is possible that fruit components are mainly composed of N, C, and O and therefore make limited contributions to soil phosphorus input. Similarly, different litter mulching treatments considerably promoted the accumulation of topsoil SOC; we observed that the content of SOC in seed mulch was the highest (47 g·kg^−1^). The fact that *C. Migao* seeds mulch the highest soil content in the topsoil may be because *C. Migao* seeds are rich in fatty acids, indicating that they are rich in carbon, and carbon can be decomposed and fed back into the soil, promoting an increase in SOC content [27,28,29]. The AN, TN, AP, TP, AK, and TK contents of soil in the pericarp and seed mulching conditions were higher than in the control. This finding indicates that seeds and pericarp are also very important for soil nutrient restoration in *C. migao* forests. The same phenomenon was observed in a study of the impact on soil nutrients of litter mulching of *Leucaena leucochala* leaves, branches, and fruits, showing that fruit decomposition is also an important factor affecting soil nutrient return [30]. The content of SOC in branch mulching was also slightly higher than that of the control. One explanation is that the decomposition rate of thin stems is slower than that of the litter from other parts of the same species, and the circulation of unstable organic matter may be much slower than that which was predicted by leaf litter data. Unstable organic matter is difficult to decompose, and the compounds enter the soil and decompose slowly after being stored in the form of organic matter [10].

Soil enzymes are very important in the litter soil feedback process. S-UE and S-ACP are significantly related to the abundance of microorganisms, and Cinnamomum migao is high in volatile oil contents, as well as litter decomposition stress microorganisms. S-CAT represents an important means for microorganisms to respond to environmental pressures with reduced stress [31]. S-INV is a hydrolase, and the enzymatic products of soil S-UE are significantly related to the content of nutrient elements (such as SOC, AN, and AP), as well as the abundance of microorganisms and the intensity of soil respiration [32]. In our study, all litter mulching treatments significantly increased the content of invertase and urease compared to the control. The change in invertase and urease activities in litter mulching was relatively consistent with the changed trend of soil nutrients under different litter mulching treatments (Figure 2). The main factor was that the abundant supply of SOC after litter mulching may have promoted the utilization of nitrogen nutrition by microbial communities, inducing soil microorganisms to increase the activity of invertase and promoting the mineralization of related compounds leached into the soil in litter [33]. This was also confirmed by the significant increase in SOC content under litter mulching compared with unmulched litter. Soil urease exhibited increased activity in litter mulching, which effectively promoted the ammoniation of N in leaching. *C. migao* litter mulching may potentially increase the supply of N and promote increased soil microbial urease activity [34,35]. In contrast, for soil catalase, the activities in the leaf, branch, and seed mulching conditions were significantly lower than those in the pericarp mulching litter condition. Interestingly, the catalase activities of soil in pericarp mulching increased sharply, whereas those of other treatments were significantly lower than those under unmulched litter, which may be due to the leaching of pericarp components into the soil. After inhibiting soil microorganisms, catalase is secreted to alleviate the damage to itself, whereas sabinene, 7 (s)-(hydroxypropanyl)-3-methyl-2-(4-oxopentyl) cyclohex-2-en-1-one, and other components in *C. migao* fruit have been shown to have obvious inhibitory effects on microorganisms [36]. However, the mulched pericarp had no significant effect on soil acid phosphatase. There was no significant difference in acid phosphatase across treatments, which may be related to the relative sufficiency of soil phosphorus content.

### 4.2. Effects of C. migao Litter Mulching on Topsoil Microbes

The niche complexity hypothesis proposes that more diverse and dispersed plant litter may promote niche specialization, which forms a variety of ecological microniches that are occupied by distinct groups of microorganisms so as to make better use of environmental sources [37]. Our study results confirm the applicability of the niche complexity hypothesis. Although Acidothermus, Sphingomonas, Burkholderia, Mortierella, Cladophialophora, and Geminibasidium were the main groups in the topsoil of the *C. migao* forest across treatments, the structure of the microbial community and the connection relationship between species changed considerably under mulching treatments with different tissues. The results confirm that diversified litter can promote the formation of soil microbial diversity (Figure 8). In the early stage of litter decomposition, the leaching of water-soluble compounds is the main process, which provides an important source of unstable carbon and nutrients for microorganisms in the underlying soil [38]. The contents of lignin and cellulose in rice leaf and pericarp are lower than those in seed and branch, making them easier to decompose. The chemical composition of different litter tissue varies greatly, and the composition (quantity and quality) of water-soluble leachate may vary considerably, which changes the soil environmental properties in different tissue mulching conditions. This change affects the microbial community structure of topsoil in different litter mulching conditions and promotes changes in the microbial ecological network [39]. In addition, macrobiological and microbiological studies have shown that the availability of resources and food is an important driving factor of biological network structure, and litter composed of different *C. migao* tissues may affect the biological network structure through input resources and food [40,41]. It is not surprising that SOC and CAT (Figure 9 and Figure 10) are the environmental factors that have the greatest impact on the microbial community under different litter mulching conditions in *C. migao* forests. Soil microbes are key factors mediating SOC decomposition. *C. migao* litter mulching with different tissues resulted in different inputs of soil SOC (Figure 3), which induced the differentiation of microbial community structure [42]. As the key enzyme used by soil microorganisms to alleviate peroxide stress in the external environment, CAT is more vulnerable to the interference of the external environment. Litter composed of different *C. migao* tissues contains different types of phenols, terpenes, olefins, and other antibacterial substances that inhibit microbes to varying degrees. Therefore, it was not surprising that soil microbial communities showed varying degrees of difference in the litter mulching treatments [36,43].

The functional preference of different microbial groups in different litter mulching treatments also indicates that litter tissue mulching resources and nutrient sources have a considerable impact on the soil microbial community (Figure 5 and Appendix A). In all treatments (including the control), the topsoil fungi were mainly saprophytic, which is in line with previous research results on the ecological function of topsoil fungi under forest litter mulching [43,44,45]. Among the fungal groups, the Zygomycota group under pericarp mulching is closely related to the release of SOC in soil, and the Chaetotyriales group under leaf litter mulching is the biocontrol microbe in soil [46]. In the branch mulching condition, as well as seed mulching and the control group, the differential fungi in topsoil, such as Mycenaceae, Remellomycetes, and Mortierella, were basically saprophytic groups and microbial groups related to cellulose degradation [47,48]. The ecological functions of soil surface bacteria showed obvious differences under different litter mulching treatments. Gemmatimonadaceae and Nitrosomonadaceae are closely related to carbon metabolism in the topsoil of unmulched litter, whereas Chryseobacterium is widespread in the soil and has strong resistance to external interference in leaf mulching [49]. In leaf mulching, Chryseobacterium, which is widespread in soil and has resistance to external interference, is the main bacterium. In branch mulching conditions, Telmatospirillum is mainly engaged in ecological functions related to lignin and cellulose decomposition, whereas Sulfuritalea is mainly engaged in the anaerobic degradation of aromatic compounds [50]. In seed mulching conditions, Protobacteria are an indicator of polycyclic aromatic hydrocarbons, which are related to the production of phosphatase by Pseudomonadales [51,52]. These findings confirm that litter composed of different *C. migao* tissues has different supplies of topsoil resources due to physical and chemical differences. A variety of chemical niches are formed in microcosm, and different microorganisms occupy different chemical niches so as to make better use of environmental resources.

## 5. Conclusions

In litter mulch obtained from different tissues of *C. miga*, changes in enzyme and nutrient content of forest soil differed according to the source tissue. In particular, pericarp and seed mulching topsoils, which has been underemphasized in previous research on plant litter soil feedback processes, contribute considerably to SOC (43.10 and 47.23 g·kg^−1^, respectively). Compared with the control, litter mulch changed the nutrient composition of surface soil, and the soil enzyme activity also changed significantly. On the other hand, as a key object of soil feedback plant litter, soil microbes also changed significantly under the action of litter, and litter mulching increased the Chao1 index diversity compared to the control. The fungal and bacterial networks covered by litter become more complex, the microbial community in pericarp mulch differed significantly from that of other treatments, and the microbial community structure was mainly affected by SOC and CAT. The above results indicate that in order to adapt to the differences of resources and food input of different tissue litter, there are considerable differences in the microbial community structure and ecological function of C. migao forest topsoil in order to promote the decomposition of litter.

## Figures and Tables

**Figure 1 microorganisms-10-01125-f001:**
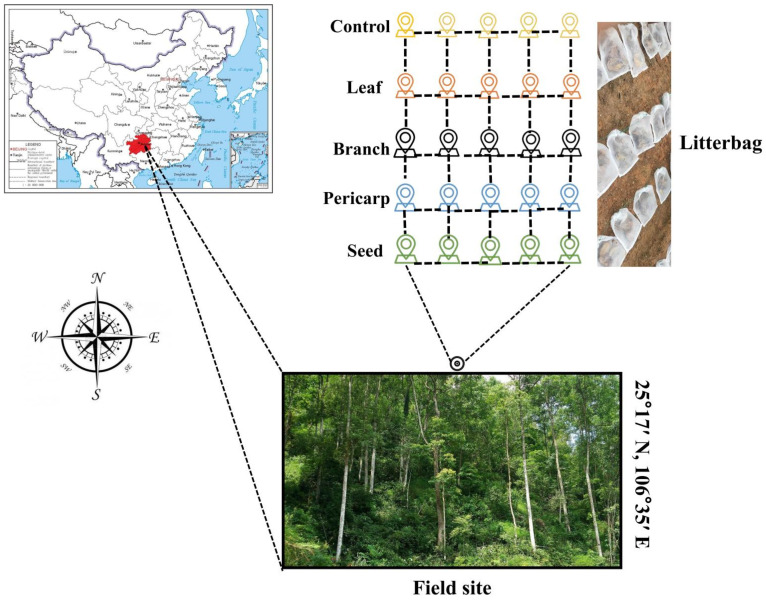
Distribution of the sampling plots and layout of the litter bags. The small legend represents litter bags placed along the sampling site.

**Figure 2 microorganisms-10-01125-f002:**
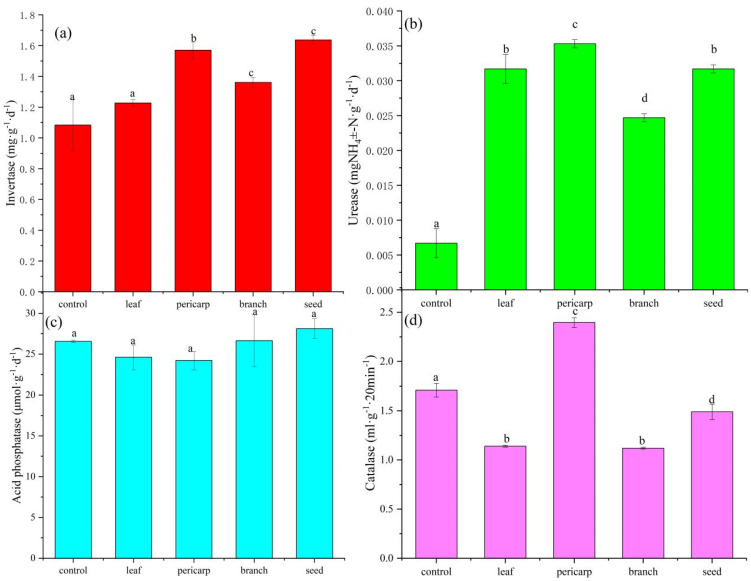
Topsoil enzyme contents in litter mulching with different *Cinnamomum migao* tissues. In a one-way ANOVA of all samples, Tukey’s test was used to compare the multiple means, and *p* < 0.05 indicated a significant difference. There is no significant difference between the same lowercase letters, and there is a significant difference between different letters. (**a**) the invertase (INV) activity, (**b**) the urease (UE) activity, (**c**) the acid phosphatase (ACP) activity, (**d**) the catalase (CAT) activity.

**Figure 3 microorganisms-10-01125-f003:**
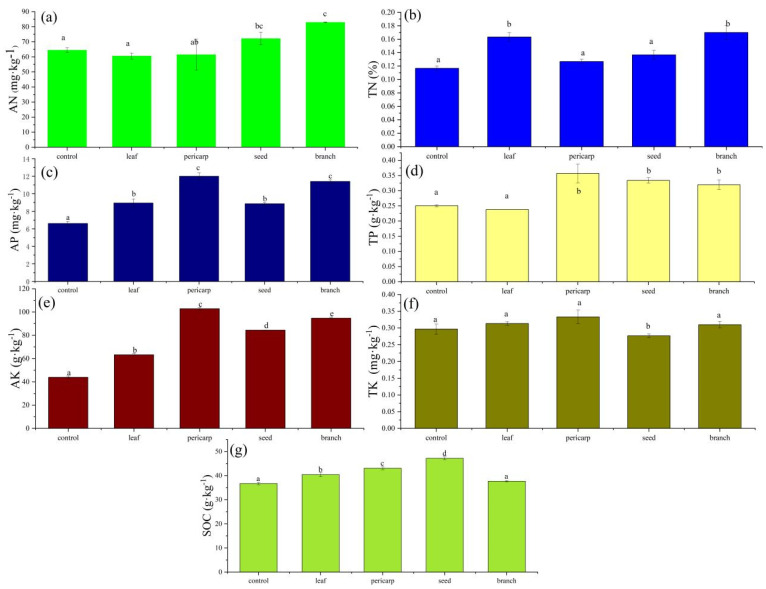
Soil nutrients in *Cinnamomum migao* litter composed of different tissues. In a one-way ANOVA of all samples, Tukey’s test was used to compare the multiple means. and *p* < 0.05 indicated a significant difference. There is no significant difference between the same lowercase letters, and there is a significant difference between different letters. (**a**) the alkali hydrogenated nitrogen (AN) content, (**b**) the total nitrogen (TN) content, (**c**) the available phosphorus (AP) content, (**d**) the total phosphorus (TP) content, (**e**) the available potassium (AK) content, (**f**) the total potassium (TK) content, (g) the soil organic carbon (SOC) content.

**Figure 4 microorganisms-10-01125-f004:**
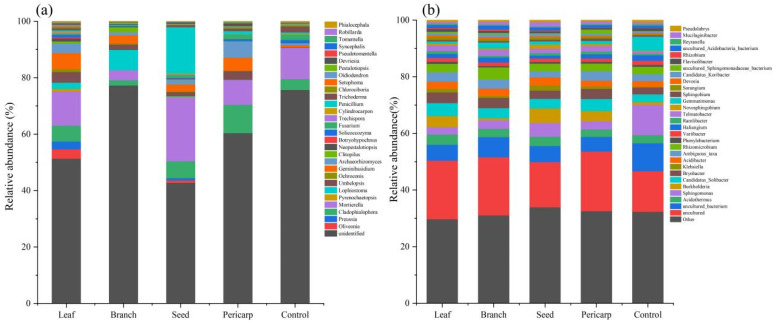
Relative abundances (%) of fungal and bacterial taxa at the genus level in topsoil under different litter mulching treatments. (**a**) Fungi; (**b**) bacteria. Mortierella, Cladophialophora, Geminibasidium, and Umbelopsis dominated the fungal microbiome in the topsoil under *C. migao* forest mulched with litter composed of different tissues (**a**). Acidothermus, Sphingomonas, and Burkholderia dominated the bacterial microbiome in the topsoil under *C. migao* forest mulched with litter composed of different tissues (**b**). Other represents the group with a relative abundance less than 1%, unidentified represents the taxonomic status of the group that cannot be defined by the database at this classification level, and uncultured represent the group cannot be cultured and for which the taxonomic status cannot be defined.

**Figure 5 microorganisms-10-01125-f005:**
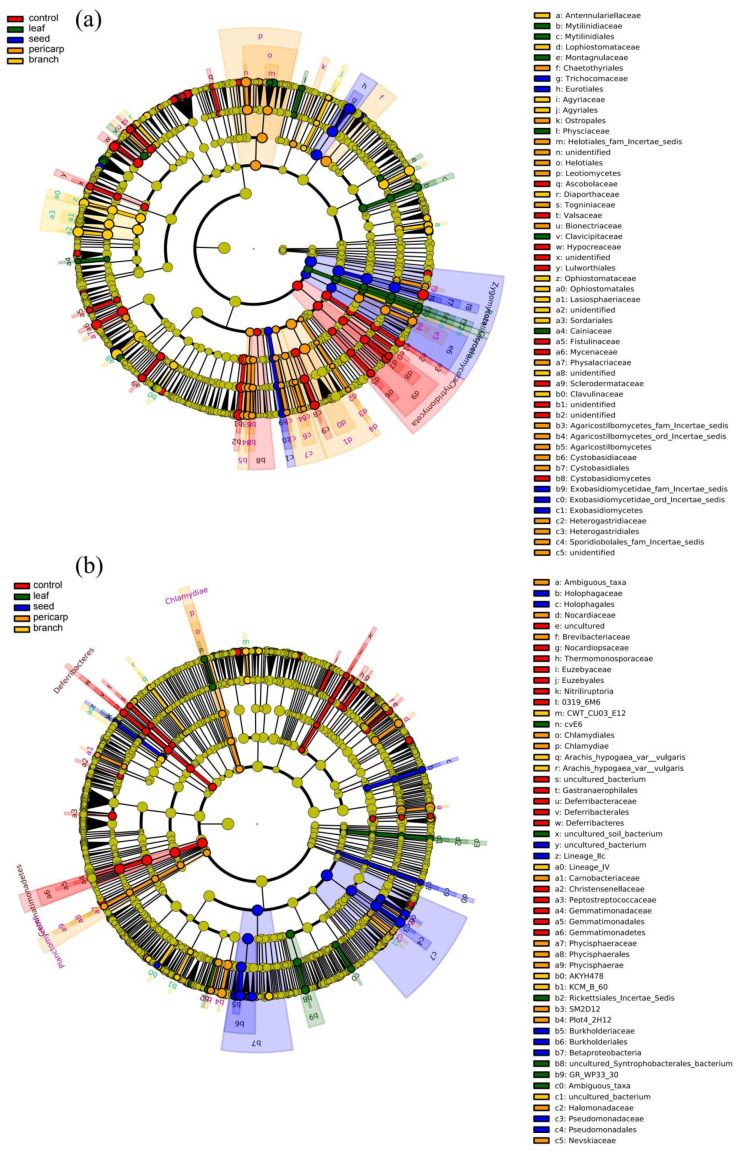
Linear discriminant analysis (LDA) coupled with effect size measurements. (**a**) Fungi, (**b**) bacteria. The circle radiating from inside to outside represents the taxonomic level from phylum to genus (or species). Each small circle at different classification levels represents a classification at that level, and the diameter of the small circle is directly proportional to the relative abundance. The species with no significant difference are uniformly colored yellow, and the different species biomarkers follow the group. The microbial groups that play an important role are in different color groups. The green node represents the microbial groups that play an important role in the green group, and the color meaning of other circles is similar. The species names represented by lowercase letters in the figure are shown in the legend on the right.

**Figure 6 microorganisms-10-01125-f006:**
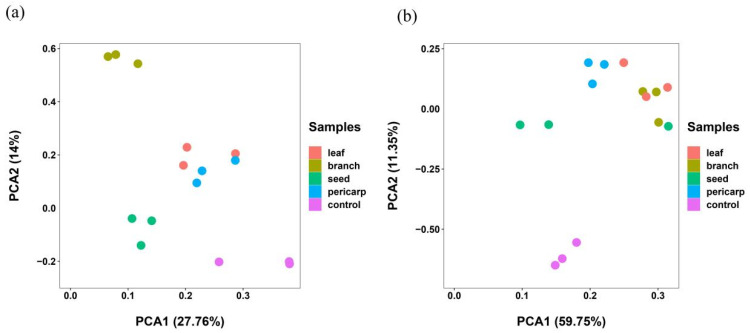
The principal component analysis (PCA) analysis was based on the operational taxonomic unit (OTU) (97% similarity clusters) abundance for soil fungi (**a**) and soil bacteria (**b**). The horizontal and vertical axes represent the two eigenvalues that can reflect the variance to the greatest extent. Each point in the figure represents a sample, the same color indicates the same group, and similar samples converge.

**Figure 7 microorganisms-10-01125-f007:**
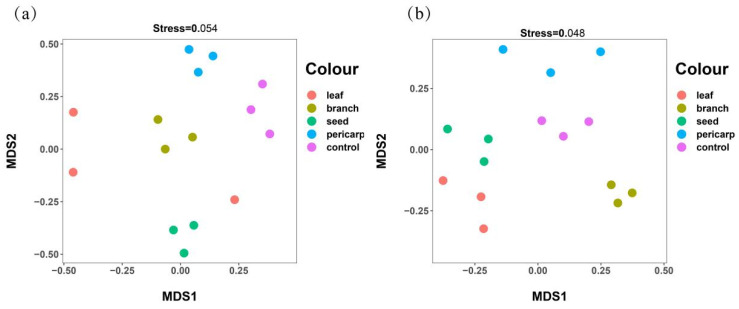
The non-metric multidimensional scaling (NMDS) analysis was based on the operational taxonomic unit (OTU) (97% similarity clusters) abundance for soil fungi (**a**) and soil bacteria (**b**). The abscissa (NMDS 1) and ordinate (NMDS 2) are two sorting axes. Each point in the figure represents a sample, the same color indicates the same group, and similar samples gather together. Large differences between samples are indicated by longer distance between points in the figure. Stress < 0.1 indicates a good ranking. Stress < 0.05 is very representative.

**Figure 8 microorganisms-10-01125-f008:**
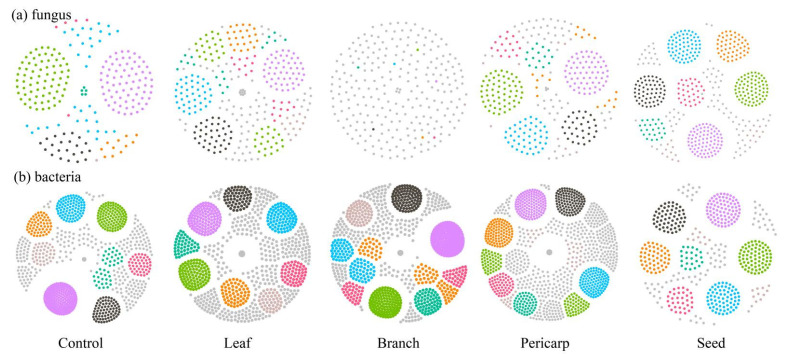
Microbial community ecological networks displayed at the operational taxonomic unit (OTU) level in topsoils in the different treatments (control, leaf, branch, pericarp, and seed). (**a**) Fungus ecological networks (OTU level), (**b**) bacteria ecological networks (OTU level).

**Figure 9 microorganisms-10-01125-f009:**
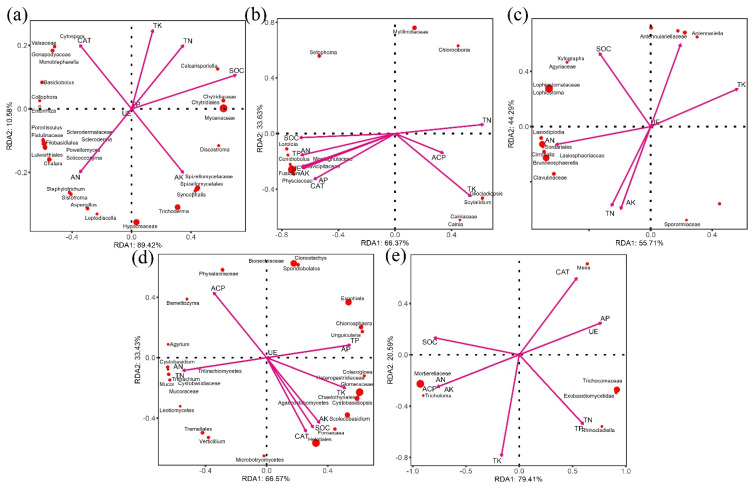
Redundancy analysis (RDA) of topsoil fungal communities under different litter mulching treatments and environmental characteristics. Arrows indicate environmental characteristics. The values of axes 1 and 2 are the percentages explained by the corresponding axis. (**a**) control, (**b**) leaf, (**c**) branch, (**d**) pericarp, (**e**) seed.

**Figure 10 microorganisms-10-01125-f010:**
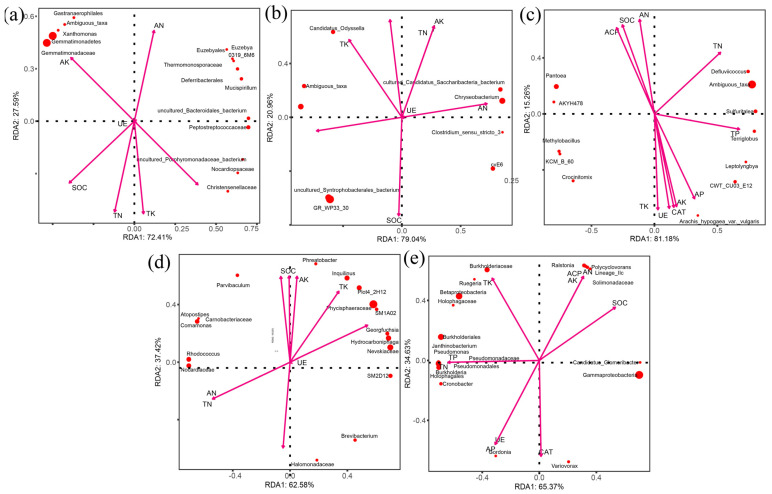
Redundancy analysis (RDA) of topsoil of bacterial communities under different litter mulching treatments and environmental characteristics. Arrows indicate environmental characteristics. The values of axes 1 and 2 are the percentages explained by the corresponding axis. (**a**) control, (**b**) leaf, (**c**) branch, (**d**) pericarp, (**e**) seed.

**Table 1 microorganisms-10-01125-t001:** Soil microbial α-diversities among the different treatments.

Different Treatment	Chao1	Shannon	Simpson
Leaf	2208.71 ± 242.922 a	8.98 ± 0.272 a	0.99 ± 0.001 a
Branch	2516.03 ± 44.451 a	9.12 ± 0.123 a	1.00 ± 0 a
Seed	2208.95 ± 380.612 a	8.43 ± 1.078 a	0.99 ± 0.015 a
Pericarp	2321.84 ± 122.665 a	8.85 ± 0.04 a	0.99 ± 0 a
Control	2159.03 ± 51.306 a	8.76 ± 0.085 a	0.99 ± 0.001 a

Results reported as the mean ± SE (*n* = 3). The same letter indicates no significant difference between groups.

**Table 2 microorganisms-10-01125-t002:** Topological characteristics of the ecological networks of the fungal communities and their associated random networks for the topsoils in different treatments.

Treatment	Ultraperipheral Node (%)		Peripheral Node (%)		Non-Hub Connectors (%)		Non-Hub Kinless Nodes (%)		Provincial Hubs (%)		Connector Hubs (%)	
Community	Fungi	Bacteria	Fungi	Bacteria	Fungi	Bacteria	Fungi	Bacteria	Fungi	Bacteria	Fungi	Bacteria
Control	52.25	0.00	41.89	38.96	0.00	30.07	3.60	30.18	2.25	0.00	0.00	0.76
Leaf	0.00	0.00	74.37	80.06	0.00	0.63	23.42	17.72	0.00	0.00	2.22	1.58
Branch	0.00	0.00	80.49	94.18	2.44	0.00	15.854	4.50	0.00	1.31	1.22	0.00
Pericarp	0.00	49.02	3.38	46.42	80.84	0.00	14.68	3.57	0.00	0.97	1.10	0.00
Seed	50.18	2.25	43.17	28.48	0	29.91	5.16	39.13	1.47	0.00	0	0.20

## Data Availability

The datasets used or analysed in the current study are available from the corresponding author on reasonable request.

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
