# Peer review of "A Case Study Demonstrates That the Litter of the Rare Species Cinnamomum migao Composed of Different Tissues Can Affect the Chemical Properties and Microbial Community Diversity in Topsoil"

_microorganisms, 2022, doi:10.3390/microorganisms10061125_

Round 1
Reviewer 1 Report
Dear Authors,
I am giving a following suggestions to help improve the quality of your manuscript:
- You need to change the topic or add to it: 'Case study....' because your research is not global only focus on an area of subtropical climate.
Additionally, you only estimated the diversity of bacteria and fungi without genes for biochemical functions, so you cannot include in the title "... microbiome structure...";
- Transfer 'soil organic carbon'(SOC)- from line 41 to line 22.
- Why were the activity of these enzymes assessed but not others, including dehydrogenases and cellulolases as well as lignolytic enzymes?
- Place in key words: ‘soil microbioms’ instid of ‘microbial’;
- Line 33: ‘…a key function.. ‘ instid of ‘… an important function…’,
- The introduction should be supplemented with information about the enzymes involved in the degradation of the tested plant tissues and their functions in the assessment of soil microbial activity.
Explain why did used soil acid phosphate (ACP), soil urease (UE),urease enzyme (UE) and invertase enzyme (INV) ?
- "2.3 Sample Collection -chapter" - no clear explanation of the aseptic sampling method for microbiological and enzymatic evaluation.
- You should put LSD in every figure and table that uses statistical analysis.
- You should explain in Figure no.4 and Fig.5 or „ Methods” ,what does means: other, unidentified, uncultured,
- 5 is illegible, you can divide it
- This manuscript has not conclusions described on the basis of the obtained test results, but the CONCLUSIONS-chapter are only a generalizations origin from the basicly of knowleadges from Soil microbiology cources.
Reviewer
Author Response
Dear reviewer
On behalf of my co-authors, we thank you very much for giving us an opportunity to revise our manuscript, we appreciate editor very much for their positive and constructive comments and suggestions on our manuscript “Litter of the rare species Cinnamomum migao composed of different tissues can affect the nutrients, microbiome structure, and ecological function of topsoil” . Thank you for pointing out many mistakes in the manuscript and giving us the opportunity to continue to improve the quality of the manuscript. According to your valuable comments, we have revised the manuscript, checked the references, and revised the questions raised, all the revise have been highlighted in red font in the text. Hopefully, our revised and answers will improve the quality of the manuscript. We also hope that the you can further point out the deficiencies of the manuscript, so that we can improve the quality of the manuscript after revise. Please see the attachment.
We will be deeply appreciated that you have done for me and look forward to your reply.
Best regards
Jiming Liu
College of Forestry, Guizhou University, Guiyang 550025, P.R.China
Tel: +86-18302618672
Email: karst0623@163.com

Reviewer 2 Report
Dear Authors,
Your manuscript is a valuable contribution with high scientific output.
I propose you to do some transformation of the text, present additional data, explanations and make a few corrections. The details are presented herewith below.
Lines 73-76
The study aims: (1) What will happen to the chemical properties of the topsoil of C. migao forest soil under litter mulching composed of different tissues? (2) Does the soil microbial community structure in the C. migao pericarp and seed mulching have the same impact as that in branch and leaf mulching?
The aims are interesting. However, whom, how and for what purpose will separate the litter in the virgin forest?
Lines 311-312
Similarly, different litter mulching treatments greatly promoted the accumulation of topsoil SOC, which was completely consistent with the current literature on litter [27-29].
This is true, but only a qualitative assessment. It should be useful to show which part of litter was mineralized in different options of you experiment. Besides, the litter is an important source of SOM.
Line 341
Niche complexity hypothesis proposes...
The hypothesis is interesting, but you addressed it only ones in the line 343.
Lines 379-380
The above results confirmed the differences in the microbial communities in the topsoil mulched with different C. migao tissues and in the unmulched treatment.
No doubts. In the lines 380-384 you have repeated the statement you presented in the lines 379-380. Please express your opinion how can be fulfilled your proposition on "better use of the resources in the environment".
Conclusion
It should be useful to give some quantitative data in the Conclusion section. High level research of the Soil Nutrient and Enzyme Concentration, Soil DNA Extraction and PCR Amplification are not presented. Please reveal what was a purpose of Sequencing Data Processing and Variance, Diversity Composition Analysis, Network Construction and Analysis, Soil microbial α-diversities among the different treatments you have provided. Many other important and interesting results you obtained but omitted in Conclusion as well.
Author Response

(The authors gave the same response as above.)
